# Full-Scale Measurements of the Propeller Thrust during Speed Trials Using Electrical and Optical Sensors

**Se-Myun Oh [1,2], Dong-Hyun Lee [1], Hyun-Joe Kim [1] and Byoung-Kwon Ahn [2,*]**

1. Samsung Heavy Industries, Daejeon 34051, Korea; semyun.oh@samsung.com (S.-M.O.); dh91.lee@samsung.com (D.-H.L.); hyunjoe.kim@samsung.com (H.-J.K.)
2. Department of Naval Architecture and Ocean Engineering, Chungnam National University, 99 Daehak-ro, Yuseong-gu, Daejeon 34134, Korea
* Correspondence: bkahn@cnu.ac.kr

**Abstract:** Full-scale sea trials demonstrate a ship's performance under real operating conditions to confirm whether a ship meets its specifications and requirements. The determination of the performance through a sea trial is the most important stage in the ship design cycle. If one is relying on measurements of propeller shaft power or fuel consumption, the distinction between the propeller and hull efficiencies may not be made. In order to be able to identify the propeller efficiency separate from the hull, full-scale propeller thrust should be accurately measured. In this study, full-scale measurements of the propeller thrust, torque, and revolution for a series of crude oil tankers and twin-skeg LNG carrier were conducted during the speed trials. Two different measuring systems, strain gauge and optical type, were implemented to compare the performance of sensors. As a result, it was shown that the strain gauge type-measuring device matched the model test results relatively well compared to the optical device. Above all, in the case of the optical device, it has been demonstrated that the zero setting is important to increase the accuracy of the full-scale measurements.

**Keywords:** ship propeller; thrust and torque; full-scale measurement; strain gauge; optical sensor; self-propulsion test; speed trial

## 1. Introduction

Recently, the interest in developing low-resistance hull form, high-efficiency propeller and energy saving devices is rapidly increasing in shipping and shipbuilding companies in accordance with rules and regulations that require minimizing fuel oil consumption and greenhouse gas emissions. In general, the speed-power performance of ships is estimated by model tests. It is believed that the evaluation of a ship's hydrodynamic performance with high accuracy and confidence level can be achieved at a model scale through towing tank experiments or numerical simulations. Then, full-scale ship performance prediction can be obtained by extrapolating the model scale results according to the friction correlation line, the law of similarity and the extrapolating method recommended by the International Towing Tank Conference (ITTC). The ITTC-recommended procedures [1] assume that the form factor is the same for the model and full-scale ship and is independent of ship speed. These assumptions have been investigated with model tests [2] and numerical analyses [3]. The scale effect of form factor depending on change in the Reynolds number has been studied and made a comparison with three kinds of friction resistance curves [4]. In order to avoid the uncertainty of scale effect, the simulation of speed-power performance at full scale is necessary [5,6]. Undoubtedly, the most advanced scenario is to simulate the self-propulsion at full scale without any assumptions, i.e., with the free surface, with the real propeller and free sink and trim [7]. The simulation results at full scale, however, must compare with scaled quantities obtained empirically according to the ITTC procedures [8].

For the substantial improvement of the hull form, propeller and energy saving devices [9], it is essential to estimate ship performance in full scale and understand the

scaling effect more deeply [10] because through the full-scale speed trials, the performance prediction from the model test can be verified [11]. In general, up to now, the speed trial has been carried out based on the propeller torque and revolution measurement [12,13]. Although a reliable measurement of the thrust at the same time allows verifying the prediction propulsive efficiency, the uncertainty of thrust measurement has been too large to obtain a reliable result to be compared with the model test prediction results [14,15]. The measurement of propeller thrust has two main benefits. Firstly, the thrust measurement gives a way to obtain speed-power performance in line with the model test. The model tests are based on the measurement of propeller revolution, torque and thrust, and the model test results are analyzed based on thrust identity. Without thrust measurement, the full-scale speed trial should be analyzed based on the torque identity. These fundamental differences can cause large errors [16]. Thus, the speed trial analysis procedure can be harmonized with the model test procedure by the thrust measurement. Moreover, the scaling effect can be understood with the measured thrust additionally [17]. Secondly, in the operation view point, the thrust measurement can reduce the operating expenditure (OPEX) through guidance on hull and propeller maintenance. The propeller efficiency can be calculated from ship speed, propeller revolution, thrust and torque. The hull resistance can be evaluated by thrust deduction fraction and propeller thrust [18]. Thus, from accurate measurements of the propeller thrust, the reduction of propeller efficiency and the increment of hull resistance can be found separately. From the monitoring of the resultant propeller efficiency and hull resistance, the time of propeller polishing and hull cleaning can be decided easily, which may save OPEX compared with the regular hull cleaning and propeller polishing by the condition-based maintenance.

In this study, to verify the results of thrust measurement using electrical and optical sensors, the relation of revolution, torque and thrust are compared with model test results. As an important index that decides the accuracy of the thrust measurement, the relation between propeller thrust and torque was investigated. In general, the relation of the propeller thrust and torque due to draught and trim conditions is rarely changed, which can be confirmed by the prediction results from model tests. In most cases, the characteristics of torque and thrust by varying engine loading show a good correlation with those of model tests. In some cases, however, the results of thrust measurement at the same torque show approximately 10% difference from the model test results, mainly due to the stability of the zero value, which is found as a problem to be solved for the stable and reliable measurement of thrust.

## 2. Measuring Devices and Systems

The shaft power meter is a device to measure torque, thrust, revolutions of propeller and power on a rotating shaft can be estimated by the measured values. Torque and thrust can be obtained by torsional and longitudinal deformation of the shaft, respectively. The torsional deformation of the shaft (usually in the order of one hundred micro strains) is generally higher than equipment tolerances (usually ten micro strains). However, the propeller thrust is the most challenging value to measure, as the longitudinal deformation of the shaft is in the same order of magnitude as the equipment tolerances [19]. Hence, even careful installation of the thrust sensor does not guarantee a high accuracy of measurements [20].

Recently, to overcome the difficulties in the thrust measurement, an optical measurement system was introduced. In this study, to identify the effect of measurement system, both the strain gauge system as a conventional system and a newly adapted optical system were installed on the propeller shaft, and the results were compared.

### 2.1. Strain Gauge-Type Measurement System

The strain gauge-type measurement system consists of a strain gauge sensor, transmitter, receiver, RPM sensor and laptop, as shown in Figure 1.

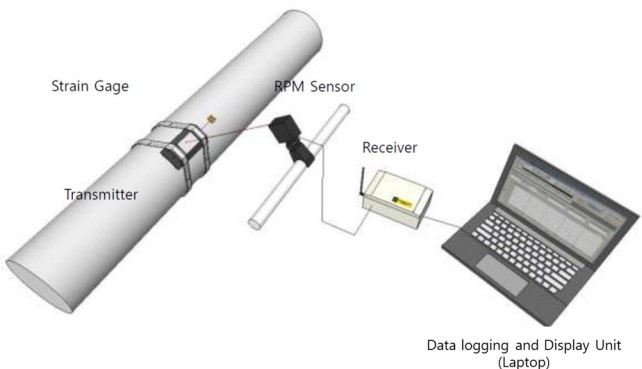

**Figure 1.** Configuration of the strain gauge-type measurement system.

The installation of the gauges on the shaft required careful treatment due to gauge's high sensitivity to the conditions of installation [21]. Therefore, before fitting the strain gauges, the sensor area on the shaft was polished well. The gauges and the amplifier were powered by batteries mounted on the shaft. For the wireless data acquisition, an antenna was also installed next to the shaft, as shown in Figure 2. To measure the shaft speed, a reflective plate was attached to the shaft line and an optical sensor was mounted close to the shaft. Each time the plate passed the sensor window a voltage signal was transmitted to a recording device, which converted to the shaft speed to rpm.

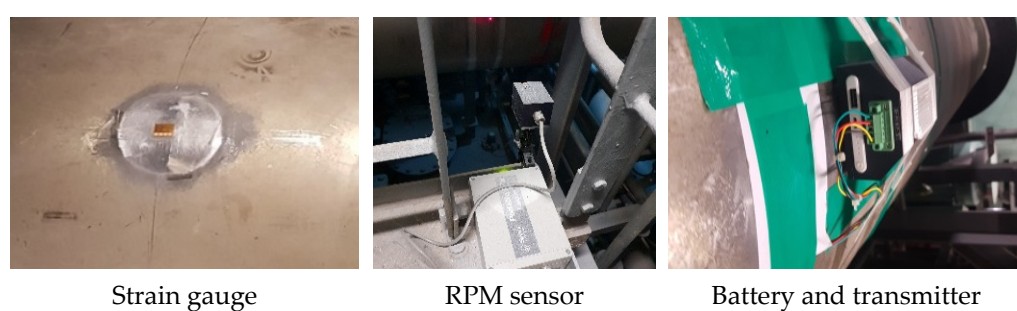

|  |  |  |
|:-:|:-:|:-:|
| Strain gauge | RPM sensor | Battery and transmitter |

**Figure 2.** Installation for the strain gauge system.

A torque measuring full-bridge arrangement with strain gauges at an angle of 45° referred to the shaft axis and the values are usually attained to a high degree of accuracy [22], with conventional thrust measuring full-bridge arrangement with strain gauges along and rectangular to the shaft axis [23], as shown in Figure 3.

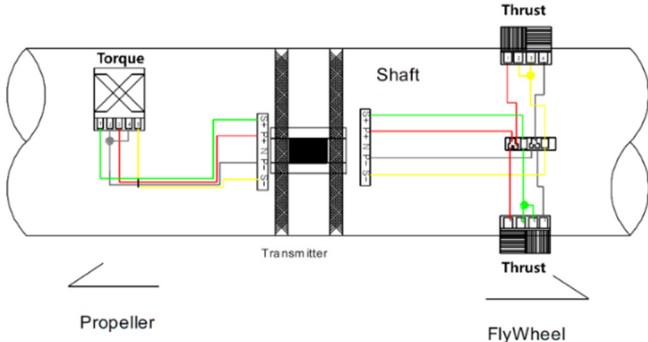

**Figure 3.** Schematic representation of strain gauges.

## 2.2. Optical-Type Measurement System

The optical shaft power meter uses LED sensors to measure shaft torque and thrust. The system consists of an LED sensor, detector arm, mounting ring and power transmission

coil on the shaft. The measured values are transferred continuously from the rotating shaft to the stator part through wireless data connection. Power transmission from the stator to the rotating shaft is performed by means of induction. The stator part consists of a power transmission coil, a data signal receiver and a control box equipped with digital output connections. The general working principle of the system [24] is shown in Figure 4.

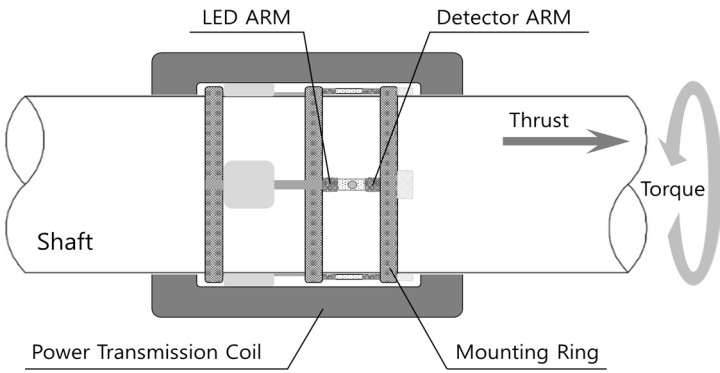

**Figure 4.** Configuration of the optical measurement system.

The sensors are mounted on propeller shafts between the propeller and the thrust bearing as shown in Figure 5. When a shaft is subject to thrust and torque, this results in a small compression and torsion of the shaft. The optical sensor measures this shaft compression and torsion over a shaft with a length of typically 200 mm. This relatively long measuring area of the shaft, compared with the conventional strain gauge-type measurement system, increases the measurement accuracy. The optical sensors detect the small displacements over the shaft length, in both axial and tangential directions, corresponding to the compression (thrust) and torsion (torque) of the propeller shaft [25].

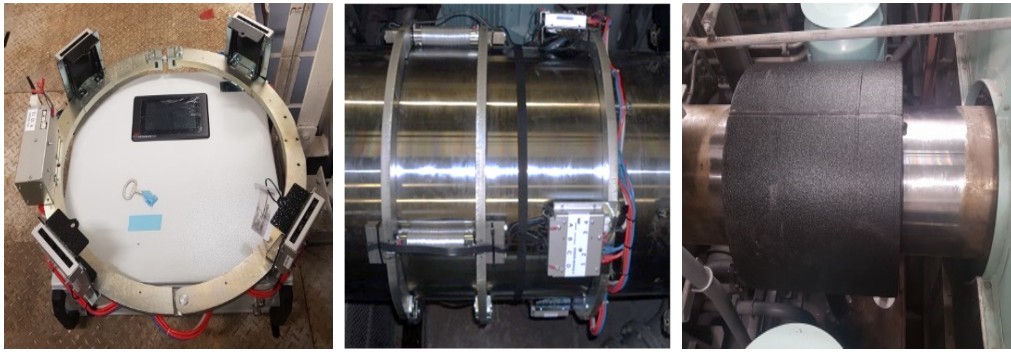

**Figure 5.** Installation for the optical measurement system.

In the optical-type measurement system, for the accurate and reliable measurement of thrust and torque, it is essential to determine the zero level very carefully. The zero setting should be performed on the condition that shaft is stationary with no residual forces or moments acting on the shaft. In this study, the zero level was adjusted after the installation during the ship was mooring on a quay side of the shipyard. Therefore, the zero setting was carried out rotating the propeller shaft clockwise and counterclockwise with the shaft revolution lower than 1 RPM to minimize the effect by the static deflection.

## 3. Results and Discussion

To identify the effects of the measurement systems, both the strain gauge and the optical-type measurement systems were installed on two vessels of a crude oil tanker series. A strain gauge system was installed on a 170,000 m$^3$ twin skeg LNG carrier, as well.

### 3.1. Crude Oil Tankers

As shown in Figure 6, the measurement systems were installed in the engine room. The strain gauge and optical sensors are mounted on propeller shafts between the propeller and the thrust bearing.

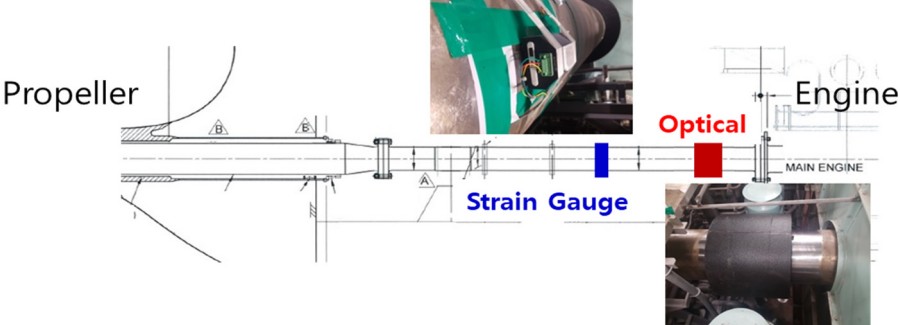

**Figure 6.** Location of the measurement systems.

The series of crude oil tanker with identical main dimensions, hull form and propulsion system were built by Samsung Heavy Industries. The speed trial of the first vessel was performed at designed draught and five sets of engine load (50%, 65%, 75%, 90%, 100% of MCR). The speed trial of the second vessel was performed at ballast draught and three sets of engine load (65%, 75%, 90% of MCR). The environmental conditions of the two vessels were moderate below Beaufort number 4. Table 1 shows the conditions of the speed trials in detail.

**Table 1.** Overview of the speed trials of crude oil tankers.

|  | **1st Vessel** | **2nd Vessel** |
|---|---|---|
| Trial Draught | Designed | Ballast |
| Beaufort No. | 4 | 3 |
| Wind Speed | 5.0~9.0 m/s | 3.0~5.0 m/s |
| Wave Height | 1.0~2.0 m | 0.5~1.0 m |
| Engine Load | 50%, 65%, 75%, 90%, 100% of MCR | 65%, 75%, 90% of MCR |

The measurement results during the speed trial of the first vessel are shown in Table 2 and Figure 7, which provide the thrust, torque and revolution of propeller measured simultaneously. The thrust and torque from measurement during speed trials, the relations of revolution–torque and revolution–thrust, are compared with prediction results based on model tests.

The characteristics of torque and thrust by the strain gauge-type system show a good correlation with those of the model test. The difference in averaged values of double runs indicates the thrust and torque are within 2.0% compared with the prediction results from model tests. The results of thrust and torque by the optical sensor show approximately 5% higher than model test and torque is 5% lower compared with the model test results.

In the optical sensor, the results of thrust and torque measurement at the 100% of MCR show different characteristics from the other engine load. At the 100% of MCR, the thrust and torque show 0.2% and 2.3% higher than prediction result from model test. In the results from the optical-type system at 100% of engine load, there is a change in the zero value, which came from the automatic zero setting function of the system in case of stopping the engine. During the sea trials, there was an event to stop the engine, and the zero levels were changed automatically by the measurement system by 40 kN of thrust and 100 Kn-m of torque from the zero values set before the test.

**Table 2.** Difference ratio between speed trial measurement and prediction results based on model test of the first vessel.

| Engine Load | | Strain Gauge (Speed Trial/Model Test) | | Optical (Speed Trial/Model Test) | |
|---|---|---|---|---|---|
| | | **Thrust** | **Torque** | **Thrust** | **Torque** |
| 65% | 1st | 99.4% | 98.4% | 104.6% | 94.0% |
| | 2nd | 101.0% | 99.3% | 106.3% | 95.1% |
| | Avg. | 100.2% | 98.9% | 105.5% | 94.6% |
| 75% | 1st | 99.5% | 98.3% | 104.8% | 94.0% |
| | 2nd | 101.1% | 99.7% | 106.6% | 95.0% |
| | Avg. | 100.3% | 99.0% | 105.7% | 94.5% |
| 75% | 3rd | 100.2% | 99.2% | 105.9% | 94.3% |
| | 4th | 98.0% | 97.7% | 104.1% | 92.8% |
| | Avg. | 99.1% | 98.5% | 105.0% | 93.6% |
| 90% | 1st | 101.3% | 99.4% | 105.9% | 95.0% |
| | 2nd | 97.1% | 96.3% | 102.5% | 92.2% |
| | Avg. | 99.2% | 97.9% | 104.2% | 93.6% |
| 100% | 1st | 104.0% | 102.2% | 102.2% | 104.1% |
| | 2nd | 99.2% | 98.3% | 98.1% | 100.4% |
| | Avg. | 101.6% | 100.3% | 100.2% | 102.3% |

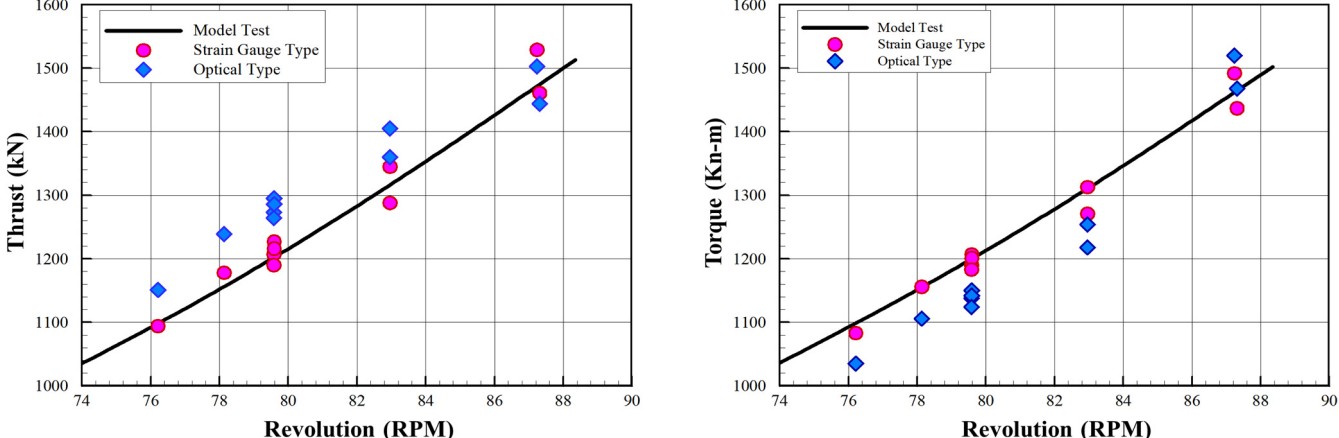

**Figure 7.** Comparison of thrust and torque between speed trials and prediction results based on model test of the first vessel.

Table 3 and Figure 8 show the results of thrust and torque measurements during speed trials of the second vessel compared with prediction results from the model test. Thrust and torque by the strain gauge measurement system are approximately 5% and 3% lower than the prediction results based on model test. In the optical sensor, the results of thrust and torque measurement at the same revolution are approximately 3% and 7% lower than the model test results.

**Table 3.** Difference ratio between speed trial measurement and prediction results based on model test of the second vessel.

| Engine Load | | Strain Gauge (Speed Trial/Model Test) | | Optical (Speed Trial/Model Test) | |
|---|---|---|---|---|---|
| | | Thrust | Torque | Thrust | Torque |
| 65% | 1st | 94.0% | 94.8% | 95.6% | 91.0% |
| | 2nd | 97.1% | 97.9% | 98.1% | 94.1% |
| | Avg. | 95.6% | 96.4% | 96.9% | 92.6% |
| 75% | 1st | 95.5% | 96.7% | 96.3% | 93.0% |
| | 2nd | 95.4% | 96.6% | 96.9% | 93.1% |
| | Avg. | 95.5% | 96.7% | 96.6% | 93.1% |
| 90% | 1st | 96.8% | 98.3% | 98.3% | 94.6% |
| | 2nd | 96.3% | 97.8% | 98.1% | 94.3% |
| | Avg. | 96.6% | 98.1% | 98.2% | 94.5% |

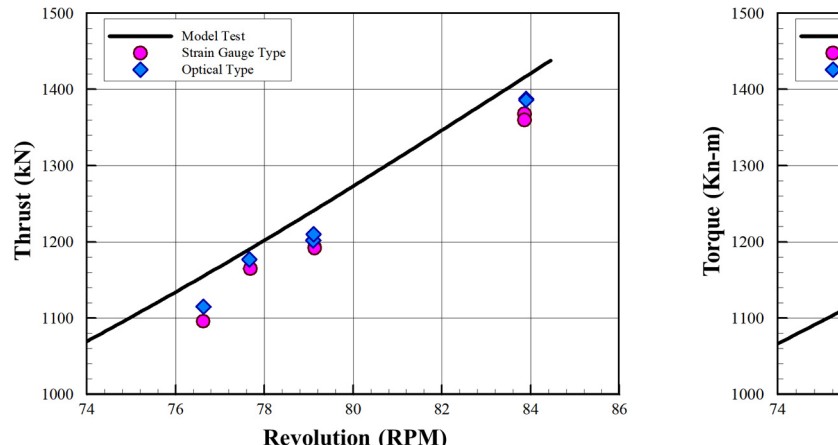 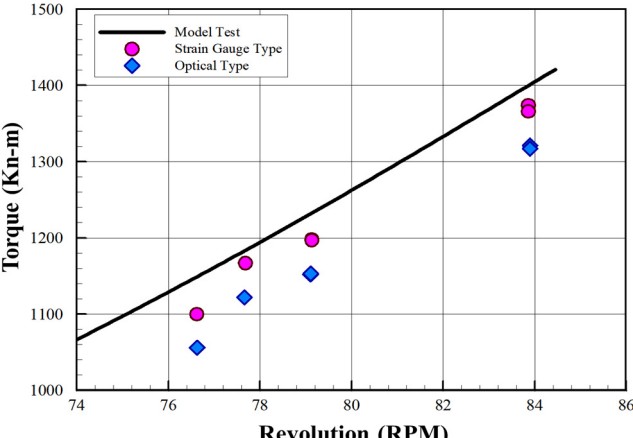

**Figure 8.** Comparison of thrust and torque between speed trials and prediction results based on model test of the second vessel.

### 3.2. Twin Skeg LNG Carrier

For the twin skeg LNG carrier, two strain gauge systems were installed on propeller shafts in the engine room at the PORT and starboard side. The speed trial was performed at ballast draught with three sets of engine load, 65%, 75%, 90% of MCR. The environmental condition was moderate below Beaufort scale 4. Table 4 shows the conditions of speed trials in detail.

The measurement results during speed trials are shown in Table 5 and Figure 9. The thrust, torque and revolution of the propeller at PORT and starboard side were measured simultaneously.

**Table 4.** Overview of the speed trials of the twin skeg LNG carrier.

| Twin Skeg LNG Carrier | |
|---|---|
| Draught | Ballast |
| Beaufort No. | 3~4 |
| Wind Speed | 3.0~10.0 m/s |
| Wave Height | 1.0~1.2 m |
| Engine Load | 65%, 75%, 90% of MCR |

**Table 5.** Difference ratio between speed trial measurement and prediction results based on the model test of the twin skeg LNG carrier.

| Engine Load | | PORT (Speed Trial/Model Test) | | Starboard (Speed Trial/Model Test) | |
|---|---|---|---|---|---|
| | | **Thrust** | **Torque** | **Thrust** | **Torque** |
| 65% | 1st | 97.1% | 96.5% | 96.1% | 97.1% |
| | 2nd | 101.3% | 100.9% | 100.5% | 101.6% |
| | Avg. | 99.2% | 98.7% | 98.3% | 99.4% |
| 75% | 1st | 94.4% | 96.0% | 93.7% | 96.3% |
| | 2nd | 98.3% | 99.2% | 97.6% | 99.9% |
| | Avg. | 96.4% | 97.6% | 95.7% | 98.1% |
| 90% | 1st | 94.2% | 96.5% | 92.9% | 96.8% |
| | 2nd | 95.6% | 97.9% | 95.6% | 98.8% |
| | Avg. | 94.9% | 97.2% | 94.3% | 97.8% |

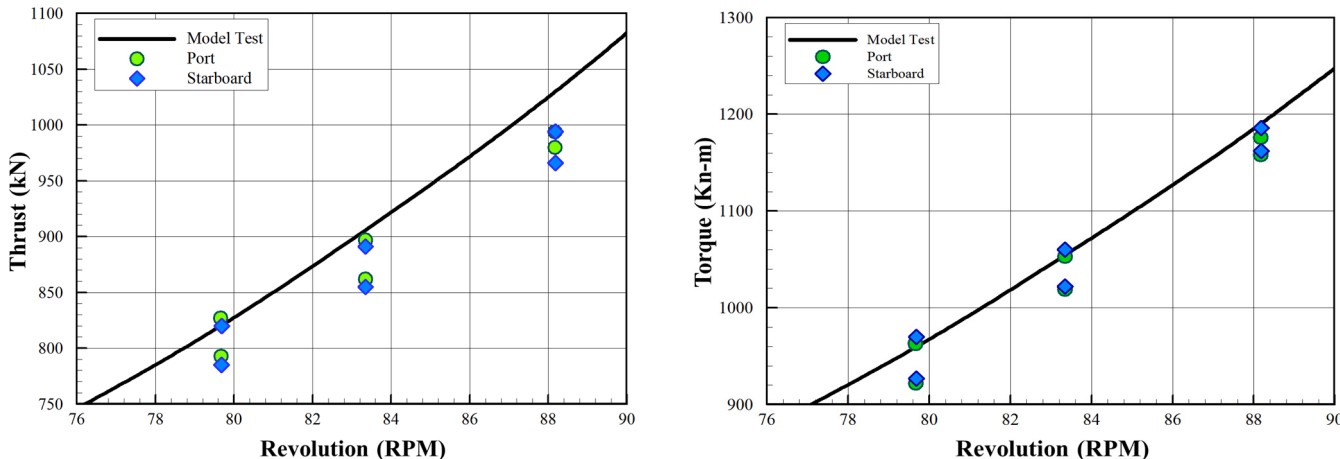

**Figure 9.** Comparison of thrust and torque between speed trials and prediction results based on the model test of the twin skeg LNG carrier.

Figure 9 presents the thrust and torque measurement results during speed trials compared with the prediction result from the model test. In the graphs, a good comparison is shown between the full-scale measurement via the torque and thrust sensor at PORT and starboard side and prediction results from model test. The results of torque measurement showed a discrepancy with model test results of approximately 2.5%. The results of thrust measurement showed the discrepancy with the model test results of 0.8% at 65% of MCR, 3.6% at 75% of MCR, and 5.7% at 100% of MCR. The comparison results of torque–thrust measurements from both sides with the prediction results show the same tendency.

The difference between the prediction results from model test and measurement during speed trial increases gradually as the engine load is increasing. This comes from the difference in the thrust measurement. The torque from the speed trial and model test agreed well with respect to the revolution of propeller. However, the measured thrust shows the differences from the prediction values from the model test, as shown in Figure 9.

### 3.3. Validation of Measurement Results

To verify the results of thrust measurement during speed trials, the relation of thrust and torque is compared with the prediction results from model test results. The relation between propeller thrust and torque is an important index that decides the accuracy of the thrust measurement and is also a good index for checking the status of the propeller. In

general, the relation of the propeller thrust and torque due to draught and trim conditions is rarely changed, which is confirmed by the model test.

Figure 10 shows the comparison result of crude oil tankers. It was found that the results of thrust measurement by the strain gauge system showed a discrepancy with thrust by the model test of approximately 2% at same torque. The characteristics of thrust–torque by varying engine loading show a good correlation with those of the model test. In cases of the optical sensor, however, the results of thrust measurement show approximately 11% at designed draught and 4% at ballast draught differences from the model test results, mainly due to the stability of the zero value.

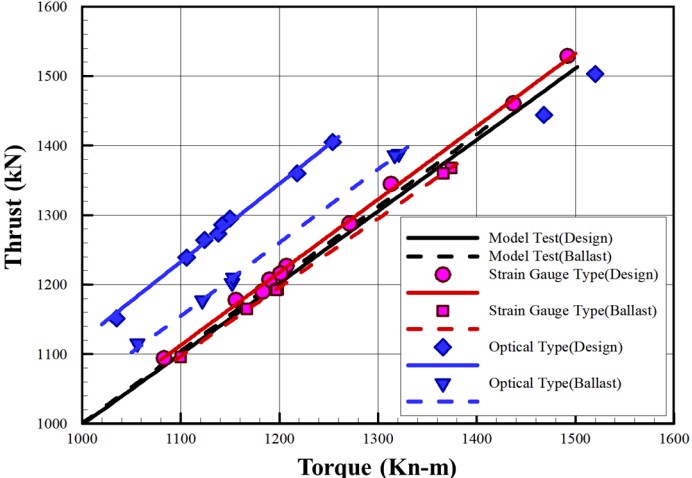

**Figure 10.** Comparison of thrust–torque of crude oil tankers.

The slopes of torque–thrust are well correlated with each other from the strain gauge system, the optical-type measurement and model tests. As an operational performance index, the characteristics of thrust and torque, such as the slope, are a main parameter. In terms of the performance index, the changes and uncertainties in the zero values might not be the issue. However, a careful posterior process should be required to obtain the proper slopes from the measurement.

Figure 11 presents the measurement results during speed trial compared with the prediction result from model test. The measurement results from both side showed the same tendency and a discrepancy with the thrust by model test of approximately 1% at same torque.

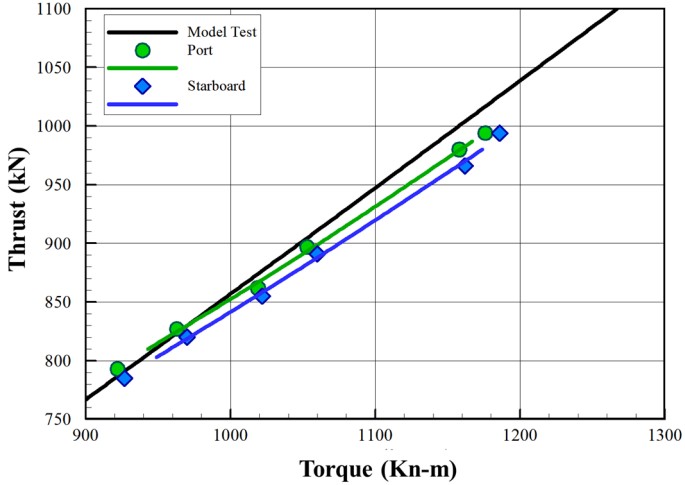

**Figure 11.** Comparison of thrust–torque of the twin skeg LNG carrier.

The difference between the prediction results from model test and measurement during speed trial increases gradually as the engine load is increasing. The results of thrust measurement at PORT side showed the discrepancy with the model test results of 0.5% at 65% of MCR, 1.3% at 75% of MCR, and 2.3% at 100% of MCR. The discrepancies at starboard side are 1.3% at 65% of MCR, 2.5% at 75% of MCR, and 3.5% at 100% of MCR. The slopes of thrust–torque are 0.903 from model tests and 0.794 from PORT and starboard side. The discrepancy of the slope is caused by the gauge factor. In general, the gauge factor is evaluated based on the shop test. For reliable results, the gauge factor should be adjusted by speed trials. The propeller thrust is a challenging value to measure. Therefore, even careful installation of a thrust measurement system does not guarantee a high accuracy of measurements.

## 4. Conclusions

Thrust measurements are a very challenging task, as the acquired axial strains are usually very small and sensitive to the zero setting of the sensor. To investigate the accuracy and reliability of propeller thrust measurements, the strain gauge-type system was installed for the twin skeg LNG carrier at PORT and starboard side, and strain gauge and optical sensor-type systems were mounted on two vessels of a crude oil tanker series. The measurement systems were installed on propeller shafts in the engine rooms between the propeller and the thrust bearing.

The summary and conclusions from this study are as follows:

Firstly, for evaluation of the thrust measurement, the relations of thrust–revolution and torque–revolution are compared with model test results for two vessels of a crude oil tanker series at designed and ballast draughts. The characteristics of torque–thrust by strain gauge show a good correlation with those of the model test. For the optical-type system, the measured thrust at the same torque show approximately 15% difference at designed draught and 5% difference at ballast draught compared with the model test results due to the uncertainty of the zero setting.

Secondly, the thrust, torque and revolution of the propeller at PORT and starboard side were measured during the speed trial of the twin skeg LNG carrier. The torque–thrust from both sides showed the same tendency. The difference between the prediction results from the model test and measurements during the speed trial increased gradually as the engine load increased. This comes from the difference in the thrust measurement. The torque from the speed trial and model test agreed well with respect to the revolution of the propeller.

Finally, the relation between propeller thrust and torque is an important index that can decide the accuracy of the thrust measurement and can also be a good index for checking the status of propellers. In general, the relation of the propeller thrust and torque due to draught and trim conditions is rarely changed, which can be confirmed by the prediction results from the model tests. As a result, the strain gauge-type sensor shows good correlations with the model test results.

The experience with the strain gauge and optical sensor shows that extensive effort is required to realize precise thrust measurement. For accurate and reliable measurements of thrust and torque, it is essential to determine the zero level carefully.

The thrust measurement can reduce the operating expenditure through guidance on hull and propeller maintenance due to the evaluation of the propeller efficiency and hull resistance. In order to monitor the power performance of ships in operation at sea, the durability of the thrust measurement system should be verified. For further study, the long-term stability of the thrust measurement will be investigated.

**Author Contributions:** Conceptualization and methodology, S.-M.O. and D.-H.L.; investigation, S.-M.O. and D.-H.L.; writing, S.-M.O.; writing—review and editing, supervision, B.-K.A.; project administration, H.-J.K. and B.-K.A. All authors have read and agreed to the published version of the manuscript.

**Funding:** This work was supported by the research fund of Chungnam National University.

**Institutional Review Board Statement:** Not applicable.

**Informed Consent Statement:** The study did not involve humans.

**Data Availability Statement:** The data presented in this study are available on request from the corresponding author.

**Acknowledgments:** All authors thank Chungnam National University for the funding.

**Conflicts of Interest:** The authors declare no conflict of interest. The funder had no role in the design of the study; in the collection, analyses or interpretation of data; in the writing of the manuscript, or in the decision to publish the results.

**Abbreviations**

The following abbreviations are used in this manuscript:

| | |
|---|---|
| ITTC | International Towing Tank Conference |
| OPEX | Operating Expenditure |
| RPM | Revolutions Per Minute |
| MCR | Maximum Continuous Rating |
| LNG | Liquefied Natural Gas |

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
