# Peer review of "Full-Scale Measurements of the Propeller Thrust during Speed Trials Using Electrical and Optical Sensors"

_applsci, doi:10.3390/app11178197_

Round 1

Reviewer 1 Report

This manuscript treats about full scale measurements of the propeller thrust during speed trials using electrical and optical sensors. The topic is interesting but the manuscript must be noticeably improved in order to publish it.

The abstract must include the main results of the work.

The introduction must be re-written. Related works with this topic must be mentioned with a critical analysis about them. Besides, the contribution of the present work to the scientific community and its novelty must be indicated and highlighted.

Section 2 must be split into measurement system and results. It is necessary to include a results section and a comprehensive analysis of the results must be developed. The results obtained are briefly explained, and this must be improved because a critical and detailed analysis of the results provides quality to any work.

Include a point at the end of lines 77, 88, 104, 114, 131, 138, 151, 153, 159, 171, 179, 198, 201, 223, and 238.

Reviewer 2 Report

The authors compare different measuring techniques to measure the thrust of a ship propeller in full scale applications. The setups of two different techniques are described and the results are compared.

Obtaining full scale measurements is difficult and the differences are large. Therefore, documenting the calibration is essential. Further, a small-scale test would allow to analyse where the differences origin from. Without in-depth analysis, one can only conclude that the uncertainty is large. Please investigate where the differences come from.

There are very few references. Please try to find more literature.

Round 2

Reviewer 1 Report

I still see a few references in the introduction and a brief description of the results. It is necessary to provide more references in the introduction section and a deeper explanation of the obtained results.

Author Response

More references in the introduction and some description of the results have been added. Please see the attachment.
